# Serum IgG titers against periodontal pathogens are associated with cerebral hemorrhage growth and 3-month outcome

**Masahiro Nakamori**[1,2], **Naohisa Hosomi**[1,3,4]*, **Hiromi Nishi**[5], **Shiro Aoki**[1], **Tomohisa Nezu**[1], **Yuji Shiga**[1], **Naoto Kinoshita**[1], **Kenichi Ishikawa**[1,2], **Eiji Imamura**[2], **Tomoaki Shintani**[6], **Hiroki Ohge**[7], **Hiroyuki Kawaguchi**[5], **Hidemi Kurihara**[8], **Shinichi Wakabayashi**[9], **Hirofumi Maruyama**[1]

1 Department of Clinical Neuroscience and Therapeutics, Hiroshima University Graduate School of Biomedical and Health Sciences, Hiroshima, Japan, 2 Department of Neurology, Suiseikai Kajikawa Hospital, Hiroshima, Japan, 3 Department of Neurology, Chikamori Hospital, Kochi, Japan, 4 Department of Disease Model, Research Institute of Radiation Biology and Medicine, Hiroshima University, Hiroshima, Japan, 5 Department of General Dentistry, Hiroshima University Hospital, Hiroshima, Japan, 6 Center of Oral Examination, Hiroshima University Hospital, Hiroshima, Japan, 7 Department of Infectious Diseases, Hiroshima University Hospital, Hiroshima, Japan, 8 Department of Periodontal Medicine, Hiroshima University Graduate School of Biomedical and Health Sciences, Hiroshima University, Hiroshima, Japan, 9 Department of Neurosurgery, Suiseikai Kajikawa Hospital, Hiroshima, Japan

* nhosomi@hiroshima-u.ac.jp

**Data Availability Statement:** All relevant data are within the Supporting Information files.

**Funding:** This study was supported by research grants from the Japan Society for the Promotion of

## Abstract

To assess the influence of periodontal disease on cerebral hemorrhage and its clinical course, we examined the association of the serum IgG titer of periodontal pathogens with hemorrhage growth and 3-month outcome. We consecutively enrolled 115 patients with acute cerebral hemorrhage (44 females, aged 71.3 ± 13.1 years) and used ELISA to evaluate the serum IgG titers of 9 periodontal pathogens: *Porphyromonas gingivalis*, *Aggregatibacter* (*A.*) *actinomycetemcomitans*, *Prevotella intermedia*, *Prevotella nigrescens*, *Fusobacterium* (*F.*) *nucleatum*, *Treponema denticola*, *Tannerella forsythensis*, *Campylobacter rectus*, and *Eikenella corrodens*. Significant hematoma growth was defined as an increase in the volume of >33% or an absolute increase in the volume of >12.5 mL. A poor outcome was defined as a 3 or higher on the modified Rankin Scale. We observed hemorrhage growth in 13 patients (11.3%). Multivariate analysis revealed that increased IgG titers of *A. actinomycetemcomitans* independently predicted the elevated hemorrhage growth (odds ratio 5.26, 95% confidence interval 1.52–18.25, *p* = 0.01). Notably, augmented IgG titers of *F. nucleatum* but not *A. actinomycetemcomitans* led to a poorer 3-month outcome (odds ratio 7.86, 95% confidence interval 1.08–57.08, *p* = 0.04). Thus, we demonstrate that elevated serum IgG titers of *A. actinomycetemcomitans* are an independent factor for predicting cerebral hemorrhage growth and that high serum IgG titers of *F. nucleatum* may predict a poor outcome in patients with this disease. Together, these novel data reveal how systemic periodontal pathogens may affect stroke patients, and, should, therefore, be taken into consideration in the management and treatment of these individuals.

Science KAKENHI (grant numbers 17K17350, 17K17907, and 18K10746) to NH. The funders had no role in study design, data collection and analysis, decision to publish, or preparation of the manuscript.

**Competing interests:** Hirofumi Maruyama reports research support from Eisai, Pfizer, Takeda Pharmaceutical, Otsuka Pharmaceutical, Nihon Pharmaceutical, Shionogi, Teijin Pharma, Fuji Film, Boehringer Ingelheim, Sumitomo Dainippon Pharma, Nihon Medi-Physics, Bayer, MSD, Daiichi Sankyo, Kyowa Hakko Kirin, Sanofi, Novartis, Kowa Pharmaceutical, Astellas Pharma, Tsumura, Japan Blood Products Organization, Mitsubishi Tanabe Pharma, and Mylan which are unrelated to the submitted work. This does not alter our adherence to PLOS ONE policies on sharing data and materials. All other authors declare that they have no conflicts of interest.

## Introduction

Periodontal disease is a risk factor for diabetes mellitus, rheumatoid arthritis, and several types of cancer [1–3]. It is a chronic inflammatory disease caused by an immune response to periodontal bacteria and characterized by loss of connective tissue and alveolar bone support, ultimately causing tooth loss [4]. Meta-analysis from previous studies showed that the risk of stroke was significantly increased in individuals with periodontitis in which the relative risk was 1.63 (95% confidence interval [CI] 1.25–2.00) [5]. Furthermore, a large cohort study has demonstrated that periodontal disease is associated with the incidence of cardioembolic and atherothrombotic stroke, and that regular dental care might decrease stroke risk [6]. Periodontitis is related to an increase in systemic inflammation markers through exposure to Gram-negative bacteria, which are implicated in the etiology of stroke [7]. While the treatment of stroke has improved remarkably, the management of periodontal disease may not only improve the clinical course in patients but could help prevent stroke altogether. However, in these studies periodontal diseases were diagnosed by only oral examination.

It was reported that serum IgG titers of a certain periodontal pathogen are considered to reflect its periodontal status [8]. Recently, by analyzing serum antibody titers, certain periodontal pathogens have been identified as risk factors for systemic diseases, such as ischemic stroke, coronary heart disease, non-alcoholic fatty liver disease, and Alzheimer's disease [9–12]. Especially regarding ischemic stroke, the serum antibody level of *Prevotella* (*P.*) *intermedia* was significantly higher in atherothrombotic stroke patients than in patients with no previous stroke [9]. However, there is no reported association between the serum IgG titers of periodontal pathogens and cerebral hemorrhage. While it was reported that serum antibody titers of *Fusobacterium* (*F.*) *nucleatum* are a predictor of unfavorable outcome after stroke [13], a thorough investigation has yet to be conducted that identifies the types of pathogens associated with stroke and its clinical course. To determine if periodontal disease affects cerebral hemorrhage, we examined the relationship among serum IgG titers of periodontal pathogens, cerebral hemorrhage growth, and a 3-month outcome.

## Materials and methods

### Patients

Consecutive acute cerebral hemorrhage patients, who were admitted to the Hiroshima University Hospital and the Suiseikai Kajikawa Hospital, Japan, from January 2013 to April 2016, were enrolled in this prospective study. The study protocols were approved by the ethics committee of the Hiroshima University Hospital (Epd-614-2) and the Suiseikai Kajikawa Hospital (2015–3) and performed according to the guidelines of the national government based on the Helsinki Declaration of 1964. Written informed consent was obtained from all patients or their relatives. All data analyses were conducted in a blinded manner.

We included patients who were admitted within 7 days from onset, were aged ≥ 20 years, and for whom consent to participate in this study was obtained from the patient or their relatives. We excluded patients who could not undergo head computed tomography (CT) and magnetic resonance imaging (MRI). We also excluded the patients who were diagnosed with hemorrhagic infarction or trauma-induced hemorrhages.

### Data acquisition

Baseline clinical characteristic data, including age, sex, body mass index (BMI), comorbidities (hypertension, diabetes mellitus, dyslipidemia, atrial fibrillation, chronic kidney disease, and stroke), smoking and drinking habits, medication of antithrombotic drugs (anti-platelet and

anticoagulant), onset to admission time, blood pressure, and C-reactive protein (CRP) levels were collected from all patients. Two stroke specialists (SA and EI) evaluated the stroke severity and conscious level. Stroke severity on admission was evaluated using the NIH Stroke Scale (NIHSS). Conscious level was evaluated with Glasgow coma scale. Comorbidities were defined according to a previous report [13] based on the Japanese hypertension, diabetes mellitus, dyslipidemia, atrial fibrillation, and chronic kidney disease guidelines. Hypertension was defined as the use of anti-hypertensive medication before admission or confirmed blood pressure of $\geq$140/90 mmHg at rest measured 2 weeks after onset. Diabetes mellitus was defined as a glycated hemoglobin level of $\geq$6.5%, fasting blood glucose level of $\geq$126 mg/dL, or use of anti-diabetes medication. Dyslipidemia was defined as a total cholesterol level of $\geq$220 mg/dL, low-density lipoprotein cholesterol level of $\geq$140 mg/dL, high-density lipoprotein cholesterol level of <40 mg/dL, triglyceride levels of $\geq$150 mg/dL, or use of anti-hyperlipidemia medication. Atrial fibrillation was defined as follows: (1) a history of sustained or paroxysmal atrial fibrillation or (2) atrial fibrillation detection on arrival or during admission. Renal functioning was calculated with the estimated glomerular filtration rate (eGFR) using a revised equation for the Japanese population as follows: eGFR (mL/min/1.73 m$^2$) = 194 $\times$ serum creatinine$^{-1.094}$ $\times$ age$^{-0.287}$ $\times$ 0.739 (for women) [14]. Chronic kidney disease was defined as an eGFR <60 mL/min/1.73 m$^2$. Imaging analysis with head computed tomography (CT) and magnetic resonance imaging (MRI) was performed in all patients for acute cerebral hemorrhage diagnosis. The cerebral hematoma was evaluated using CT. The admission and follow-up CT scans (24 hours after admission) were performed with axial 5-mm section thickness. Two experienced neurologists (MN and YS) measured the cerebral hematomas using the ABC/2 formula. The neuroimaging evaluation remained blinded from the clinical assessment. Based on the criteria used in several large clinical studies, hematoma growth was defined as an increase in hematoma volume of >33% or >12.5 mL at 24-hour follow-up [15]. The other two experienced neurologists (NK and KI) also performed evaluations of MRI findings and hematoma growth neuroimaging predictors, which were detected as the blend sign or black hole sign using plain head CT [15]. Cerebral amyloid angiopathy (CAA) was diagnosed using modified Boston criteria [16].

We collected clinical data regarding the acute phase, including intraventricular hemorrhage extension, pharmacological blood pressure management during the first 24 hours, surgical management approach, tube ventilatory use, and septic complications.

When we evaluated the 3-month outcome, we excluded patients who were disabled prior to stroke (corresponding to premorbid modified Rankin Scale [mRS] score $\geq$2). An unfavorable 3-month outcome was defined as a 3 or higher on the mRS.

## Measurement of serum antibody titers of periodontal pathogens

Serum IgG antibody titers of periodontal pathogens were determined using ELISA as previously described [11]. Bacterial antigen-coated wells were washed with phosphate-buffered saline with Tween (PBST); serum samples in PBST were then added to the wells. After incubation at 4˚C overnight, the wells were washed with PBST and filled with alkaline phosphatase-conjugated goat anti-human IgG (gamma-chain specific, Abcam, Cambridge, MO) in PBST. After another incubation at 37˚C for 2 hours, the wells were again washed with PBST, an aliquot of p-nitrophenylphosphate at 1 mg/mL (WAKO Pure Chemical Industries Ltd., Osaka, Japan) in 10% diethanolamine buffer was added to each well as a substrate, and incubation was performed at 37˚C for 30 minutes. Optical density at 405 nm was measured using a microplate reader (iMark, Bio-Rad Laboratories Inc., Hercules, CA). Serum samples were collected from patients within 3 days after stroke onset and stored at −80˚C. Sonicated preparations of the following periodontal pathogens were used as bacterial antigens *Porphyromonas* (P.) *gingivalis*,

*Aggregatibacter* (*A.*) *actinomycetemcomitans*, *P. intermedia*, *Prevotella nigrescens*, *F. nucleatum*, *Treponema denticola*, *Tannerella forsythensis*, *Campylobacter rectus*, and *Eikenella corrodens*. We selected these representative periodontal pathogens based on the previously reported association of serum antibody titers with stroke outcome [13]. The serum of 5 healthy individuals was pooled and used for calibration. Using serial dilutions of the pooled control serum, the standard reaction was defined based on the ELISA unit (EU) such that 100 EU corresponded to a 1:3200 dilution of the calibrator sample. For statistical analysis, we used the common logarithms of serum IgG antibody titers.

## Statistical analysis

Data are expressed as the mean ± standard deviation or the median (minimum, maximum) for continuous variables, and frequencies and percentages for discrete variables. Statistical analysis was performed using the JMP 14.0 statistical software (SAS Institute Inc., Cary, NC, USA). The statistical significance of intergroup differences was assessed using the unpaired *t*-test or Mann-Whitney *U* test for continuous variables or the Fisher exact test or $\chi^2$ test for discrete variables as appropriate. Because there were no reports of hemorrhagic stroke, we calculated the sample size according to the past investigations for the IgG titers of periodontal pathogens in atherothrombotic stroke [9]. Based on an alpha level = 0.05 and power = 0.80, we estimated that we would require a total of n = 99 participants. Baseline data of cerebral hemorrhage patients were analyzed, and two-step strategies were employed to assess the relative importance of variables in their association with hemorrhage growth and poor outcome using least square linear regression analysis. We first performed a univariate analysis, followed by a multi-factorial analysis with selected factors with $p < 0.05$ in the former analysis. We considered $p < 0.05$ as statistically significant. We also calculated the statistical power and effect size as post hoc analysis.

## Results

A total of 115 patients (44 females, aged 71.3 ± 13.1 years) with acute cerebral hemorrhage were registered in this study. The baseline clinical characteristics are shown in Table 1. Among them, 32 (27.8%) patients had histories of stroke, 16 of whom were cerebral hemorrhage. The number of patients with anti-platelet and anticoagulant drug use was 20 (17.4%) and 15 (13.2%), respectively. The median time from onset to admission was 147 minutes (min–max: 26–7200). CAA was diagnosed 16 (13.9%) patients, of whom 7 patients were probable CAA and 6 patients were possible CAA. The mean serum IgG titers of periodontal disease pathogens from all patients are summarized in Table 2.

We found hemorrhage growth in 13 patients (11.3%). The potential factors associated with hemorrhage growth were evaluated using the univariate analysis (listed in Tables 1 and 2). In this analysis, hemorrhage growth was associated with the history of atrial fibrillation, usage of anticoagulant, and the IgG titers of *A. actinomycetemcomitans*. Among patients with hemorrhage growth, 5 (38.5%) patients had used an anticoagulant, namely warfarin, and were not injected with the prothrombin complex concentrate during the time of admission due to the Japanese medical insurance system at that time. Multivariate logistic regression analysis revealed that usage of anticoagulant (odds ratio 8.36, 95% CI 1.35–51.70, $p = 0.02$), septic complications (odds ratio 10.20, 95% CI 1.94–53.72, $p = 0.01$), and the IgG titer of *A. actinomycetemcomitans* (odds ratio 5.26, 95% CI 1.52–18.25, $p = 0.01$) were independently associated with hemorrhage growth (Table 3). Regarding the IgG titer of *A. actinomycetemcomitans*, the statistical power and effect size of Cohen's *d* were 0.80 and 0.67, respectively.

**Table 1. Baseline clinical characteristics.**

|  | n = 115 |
|---|---|
| Age, mean±SD | 71.3±13.1 |
| Sex [female], n (%) | 44 (38.3) |
| Body mass index, kg/m$^2$, mean±SD | 22.3±4.1 |
| History |  |
| Hypertension, n (%) | 96 (83.5) |
| Diabetes mellitus, n (%) | 21 (18.3) |
| Dyslipidemia, n (%) | 30 (26.1) |
| Atrial fibrillation, n (%) | 13 (11.3) |
| Chronic kidney disease, n (%) | 35 (30.4) |
| Stroke, n (%) | 32 (27.8) |
| cerebral hemorrhage, n (%) | 16 (50.0)[*] |
| Duration from the past stroke, month, median (minimum, maximum) | 36 (0.67, 384)[*] |
| Current smoker, n (%) | 52 (47.3) |
| Habitual drinker, n (%) | 30 (27.3) |
| Usage of anti-platelet drug, n (%) | 20 (17.4) |
| Usage of anticoagulant drug, n (%) | 15 (13.2) |
| Time from onset to admission, minutes, median (minimum, maximum) | 147 (26, 7200) |
| Systolic blood pressure on admission, mmHg, mean±SD | 175.8±30.8 |
| Diastolic blood pressure on admission, mmHg, mean±SD | 100.0±17.9 |
| NIHSS score, median (minimum, maximum) | 9 (0, 38) |
| Glasgow coma scale, median (minimum, maximum) | 15 (4, 15) |
| CRP, mg/dl, mean±SD | 0.75±1.31 |
| Premorbid mRS ≥2, n (%) | 22 (19.1) |
| Cerebral hematoma on admission |  |
| Volume, ml, mean±SD | 15.5±20.6 |
| Supratentorial, n (%) | 94 (81.7) |
| intraventricular hemorrhage extension, n (%) | 38 (33.0) |
| hematoma growth neuroimaging predictors, n (%) | 13 (11.3) |
| Pharmacological blood pressure management during the first 24 hours, n (%) | 92 (80.0) |
| Surgical Management approach, n (%) | 9 (7.8) |
| Tube ventilatory use, n (%) | 9 (7.8) |
| In hospital septic complications, n (%) | 14 (12.2) |
| Etiology |  |
| Hypertensive, n (%) | 92 (80.0) |
| Cerebral amyloid angiopathy, n (%) | 16 (13.9) |
| Others, n (%) | 7 (6.1) |

SD, standard deviation; NIHSS, NIH Stroke Scale; CRP, C-reactive protein; mRS, modified Rankin Scale.
[*]n = 32.

When we evaluated the 3-month outcome, 22 patients were excluded based on premorbid mRS scores of ≥2. The potential factors associated with poor outcome (mRS score ≥3) (listed in Tables 1–3) were evaluated using the univariate analysis. We found that poor outcome was associated with age, NIHSS score, cerebral hematoma volume, cerebral hematoma growth, and IgG titers of *F. nucleatum*. Multivariate logistic regression analysis revealed that age (odds ratio 1.09, 95% CI 1.01–1.17, *p* = 0.02), NIHSS score (odds ratio 1.29, 95% CI 1.09–1.52, *p* = 0.002), and the IgG titer of *F. nucleatum* (odds ratio 7.86, 95% CI 1.08–57.08, *p* = 0.04)

**Table 2. Mean serum IgG titers of periodontal disease pathogens in all cerebral hemorrhage patients.**

|  | IgG titer (EU) |
| --- | --- |
| *Porphyromonas gingivalis* | 656.0 ± 1949.9 |
| *Aggregatibacter actinomycetemcomitans* | 185.1 ± 454.2 |
| *Prevotella intermedia* | 131.6 ± 191.1 |
| *Prevotella nigrescens* | 74.3 ± 111.0 |
| *Fusobacterium nucleatum* | 54.8 ± 87.1 |
| *Treponema denticola* | 43.0 ± 88.0 |
| *Tannerella forsythensis* | 41.3 ± 88.7 |
| *Campylobacter rectus* | 201.2 ± 511.3 |
| *Eikenella corrodens* | 61.1 ± 92.4 |

EU, ELISA unit.

were independently associated with poor outcome, but not cerebral hematoma volume or growth (Table 4). Regarding the IgG titer of *F. nucleatum*, the statistical power and effect size of Cohen's *d* were 0.91 and 0.69, respectively.

## Discussion

Our findings suggest that periodontal disease may be associated with cerebral hemorrhage and its clinical course. Specifically, we reveal for the first time that serum IgG titers of a few particular periodontal pathogens may be useful biomarkers for predicting the clinical course of a cerebral hemorrhage.

In the present study, we provide evidence that an elevated serum IgG titer of *A. actinomycetemcomitans* is an independent factor for predicting hemorrhage growth. *A. actinomycetemcomitans* is a gram-negative, facultatively anaerobic coccobacillus and is considered the major etiologic agent of localized aggressive periodontitis [17]. It also contributes to chronic periodontitis, and according to previous reports, an elevated serum titer of *A. actinomycetemcomitans* predicts stroke and coronary heart disease [18–20]. A potential explanation for these relationships is that individuals infected with *A. actinomycetemcomitans* harbor T-cells specific to this bacterium in their blood, which express the receptor activator of nuclear factor-κB (RANK) ligand [21]. This ligand stimulates the vascular smooth muscle cells to produce matrix metalloproteinase-9, which may promote the growth of cerebral hemorrhages.

We further demonstrate that the serum IgG titer of *F. nucleatum* independently predicted an unfavorable outcome in cerebral hemorrhage, which is consistent with a previous study [13]. *Fusobacterium nucleatum* is strictly an anaerobic gram-negative rod bacterium, normally found in the oral cavity. It is considered to be a periodontal pathogen because it is frequently isolated from periodontitis lesions, produces a high number of tissue irritants, and often aggregates with other periodontal pathogens as a bridge between early and late colonizers [22,23]. *Fusobacterium nucleatum*, which can pass the blood-brain barrier, is associated with colon cancer and Alzheimer's disease [3,12], and was found to cause brain abscesses [24,25]. This bacterium also has abilities to adhere to and invade host vascular endothelial cells using the protein, Fusobacterium adhesin A (FadA), which binds to vascular endothelial-cadherin on the cell surface, thereby triggering a breakdown of endothelial cell-to-cell junctions [26]. The virulence of *F. nucleatum* mediates endotoxin activity, hemagglutination, as well as aggregation and death of immune cells via their outer membrane proteins, such as fibroblast activation protein 2 (FAP-2) and radiation genes (RadD) [27]. In addition, *F. nucleatum* causes an increase in the expression of genes associated with host immune responses, such as

**Table 3. Factors influencing cerebral hemorrhage growth.**

| | Hematoma growth | | Univariate analysis | Multivariate analysis | | |
|---|---|---|---|---|---|---|
| | (+) n = 13 | (-) n = 102 | p value | odds ratio | 95% CI | p value |
| Age, mean±SD | 74.0±7.8 | 71.0±13.6 | 0.43 | | | |
| Sex [female], n (%) | 3 (23.1) | 41 (40.2) | 0.23 | | | |
| Body mass index, kg/m², mean±SD | 23.3±5.3 | 22.2±3.9 | 0.37 | | | |
| Hypertension, n (%) | 11 (84.6) | 85 (83.3) | 0.91 | | | |
| Diabetes mellitus, n (%) | 3 (23.1) | 18 (17.7) | 0.63 | | | |
| Dyslipidemia, n (%) | 4 (30.8) | 26 (26.5) | 0.68 | | | |
| Atrial fibrillation, n (%) | 4 (30.8) | 9 (8.8) | 0.02* | 1.69 | 0.28–10.34 | 0.57 |
| Chronic kidney disease, n (%) | 5 (38.5) | 30 (29.4) | 0.50 | | | |
| Stroke, n (%) | 6 (46.2) | 26 (25.5) | 0.12 | | | |
| Duration from the past stroke, n (%) | 22 (12, 384) | 60 (0.67, 312) | 0.33 | | | |
| Current smoker, n (%) | 5 (38.5) | 47 (48.5) | 0.50 | | | |
| Habitual drinker, n (%) | 3 (23.1) | 27 (27.8) | 0.72 | | | |
| Usage of anti-platelet, n (%) | 1 (7.7) | 19 (18.6) | 0.33 | | | |
| Usage of anticoagulant, n (%) | 5 (38.5) | 10 (9.9) | 0.004* | 8.36 | 1.35–51.70 | 0.02* |
| Time from onset to admission, minute, median (minimum, maximum) | 240 (69, 5760) | 135.5 (26, 7200) | 0.11 | | | |
| Systolic blood pressure on admission, mmHg, mean±SD | 175.9±22.3 | 175.8±31.8 | 0.99 | | | |
| Diastolic blood pressure on admission, mmHg, mean±SD | 99.8±13.3 | 100.1±18.4 | 0.97 | | | |
| NIHSS score, median (minimum, maximum) | 16 (0, 38) | 9 (0, 38) | 0.17 | | | |
| Glasgow coma scale, median (minimum, maximum) | 14 (4, 15) | 15 (4, 15) | 0.33 | | | |
| CRP, mg/dl, mean±SD | 1.30±1.94 | 0.68±1.21 | 0.11 | | | |
| Cerebral hematoma volume on admission, ml, mean±SD | 9.47±8.26 | 16.29±21.53 | 0.26 | | | |
| Supratentorial hematoma, n (%) | 10 (76.9) | 84 (82.4) | 0.63 | | | |
| Intraventricular hemorrhage extension, n (%) | 7 (53.9) | 31 (30.4) | 0.09 | | | |
| Hematoma growth neuroimaging predictors, n (%) | 1 (7.7) | 12 (11.8) | 0.66 | | | |
| Pharmacological blood pressure management during the first 24 hours, n (%) | 13 (100.0) | 79 (77.5) | 0.06 | | | |
| Surgical Management approach, n (%) | 1 (7.7) | 8 (7.8) | 0.98 | | | |
| Tube ventilatory use, n (%) | 2 (15.4) | 7 (6.9) | 0.28 | | | |
| In hospital septic complications, n (%) | 4 (30.8) | 10 (9.8) | 0.03* | 10.2 | 1.94–53.72 | 0.01* |
| Hypertensive cerebral hemorrhage, n (%) | 12 (92.3) | 80 (78.4) | 0.24 | | | |
| Cerebral amyloid angiopathy, n (%) | 0 (0) | 16 (15.7) | 0.12 | | | |
| Other etiologies, n (%) | 1 (7.7) | 6 (5.9) | 0.80 | | | |
| IgG titer of periodontal disease pathogen | | | | | | |
| log P. gingivalis, mean±SD | 2.10±1.03 | 2.01±0.91 | 0.75 | | | |
| log A. actinomycetemcomitans, mean±SD | 2.15±0.78 | 1.69±0.67 | 0.02* | 5.26 | 1.52–18.25 | 0.01* |
| log P. intermedia, mean±SD | 2.07±0.73 | 1.76±0.58 | 0.08 | | | |
| log P. nigrescens, mean±SD | 1.48±0.87 | 1.45±0.73 | 0.91 | | | |
| log F. nucleatum, mean±SD | 1.62±0.61 | 1.39±0.54 | 0.21 | | | |
| log T. denticola, mean±SD | 1.30±0.46 | 1.10±0.75 | 0.34 | | | |
| log T. forsythensis, mean±SD | 1.38±0.52 | 1.23±0.60 | 0.38 | | | |
| log C. rectus, mean±SD | 1.92±0.75 | 1.71±0.70 | 0.31 | | | |
| log E. corrodens, mean±SD | 1.73±0.50 | 1.45±0.59 | 0.11 | | | |

CI, confidence interval; SD, standard deviation; NIHSS, NIH Stroke Scale; CRP, C-reactive protein.

* means $p < 0.05$.

**Table 4. Factors influencing unfavorable outcome (modified Rankin Scale ≥3).**

| | Outcome | | Univariate analysis | Multivariate analysis | | |
|---|---|---|---|---|---|---|
| | Favorable, n = 48 | Unfavorable, n = 45 | p value | odds ratio | 95% CI | p value |
| Age, mean±SD | 66.9±12.1 | 73.4±11.5 | 0.01* | 1.09 | 1.01–1.17 | 0.02* |
| Sex(female), n (%) | 14 (29.2) | 19 (42.2) | 0.23 | | | |
| Body mass index, kg/m$^2$, mean±SD | 22.8±4.0 | 22.6±4.4 | 0.83 | | | |
| Hypertension, n (%) | 42 (87.5) | 37 (82.2) | 0.48 | | | |
| Diabetes mellitus, n (%) | 7 (14.6) | 10 (22.2) | 0.34 | | | |
| Dyslipidemia, n (%) | 16 (30.1) | 12 (26.7) | 0.48 | | | |
| Atrial fibrillation, n (%) | 5 (10.4) | 5 (11.1) | 0.91 | | | |
| Stroke, n (%) | 8 (16.7) | 10 (22.2) | 0.50 | | | |
| Duration from the past stroke, month, median (minimum, maximum) | 54 (12, 312) | 60 (0.8, 384) | 0.70 | | | |
| Current smoker, n (%) | 28 (59.6) | 17 (41.5) | 0.09 | | | |
| Habitual drinker, n (%) | 16 (34.0) | 10 (24.4) | 0.32 | | | |
| Usage of anti-platelet drug, n (%) | 8 (16.7) | 6 (13.3) | 0.65 | | | |
| Usage of anticoagulant, n (%) | 4 (8.3) | 7 (15.6) | 0.28 | | | |
| Time from onset to admission, minute, median (minimum, maximum) | 143.5 (26, 7200) | 153 (29, 5760) | 0.95 | | | |
| Systolic blood pressure on admission, mmHg, mean±SD | 177.3±31.0 | 175.3±30.2 | 0.75 | | | |
| Diastolic blood pressure on admission, mmHg, mean±SD | 100.7±18.5 | 100.5±17.4 | 0.95 | | | |
| NIHSS score, median (minimum, maximum) | 3 (0, 19) | 14 (0, 29) | <0.001* | 1.29 | 1.09–1.52 | 0.002* |
| Glasgow coma scale, median (minimum, maximum) | 15 (11, 15) | 14 (4, 15) | <0.001* | 1.57 | 0.98–2.53 | 0.06 |
| CRP, mg/dl, mean±SD | 0.50±1.00 | 0.97±1.63 | 0.09 | | | |
| Cerebral hematoma volume on admission, ml, mean±SD | 6.63±9.36 | 19.75±29.21 | <0.001* | 1.00 | 0.93–1.08 | 0.90 |
| Cerebral hematoma growth, n (%) | 1 (2.1) | 9 (20.0) | 0.01* | 14.08 | 0.98–202.21 | 0.05 |
| Supratentorial hematoma, n (%) | 36 (75.0) | 38 (84.4) | 0.26 | | | |
| Intraventricular hemorrhage extension, n (%) | 5 (10.4) | 21 (46.7) | <0.001* | 5.17 | 0.93–28.67 | 0.06 |
| hematoma growth neuroimaging predictors, n (%) | 2 (4.2) | 5 (11.1) | 0.20 | | | |
| Pharmacological blood pressure management during the first 24 hours, n (%) | 34 (70.8) | 39 (86.7) | 0.06 | | | |
| Surgical Management approach, n (%) | 1 (2.1) | 5 (11.1) | 0.08 | | | |
| Tube ventilatory use, n (%) | 1 (2.1) | 5 (11.1) | 0.08 | | | |
| In hospital septic complications, n (%) | 1 (2.1) | 8 (17.8) | 0.01* | 8.72 | 0.47–162.50 | 0.15 |
| Hypertensive cerebral hemorrhage, n (%) | 37 (77.1) | 38 (84.4) | 0.37 | | | |
| Cerebral amyloid angiopathy, n (%) | 8 (16.7) | 4 (8.9) | 0.26 | | | |
| Other etiologies, n (%) | 3 (6.3) | 3 (6.7) | 0.93 | | | |
| IgG titer of periodontal disease pathogen | | | | | | |
| log P. gingivalis, mean±SD | 1.86±1.01 | 2.20±0.74 | 0.07 | | | |
| log A. actinomycetemcomitans, mean±SD | 1.78±0.81 | 1.73±0.64 | 0.73 | | | |
| log P. intermedia, mean±SD | 1.69±0.68 | 1.86±0.55 | 0.19 | | | |
| log P. nigrescens, mean±SD | 1.44±0.82 | 1.50±0.72 | 0.74 | | | |
| log F. nucleatum, mean±SD | 1.28±0.48 | 1.63±0.51 | <0.001* | 7.86 | 1.08–57.08 | 0.04* |
| log T. denticola, mean±SD | 1.11±0.75 | 1.21±0.67 | 0.47 | | | |
| log T. forsythensis, mean±SD | 1.29±0.54 | 1.25±0.62 | 0.74 | | | |
| log C. rectus, mean±SD | 1.70±0.67 | 1.78±0.74 | 0.57 | | | |

(*Continued*)

**Table 4.** (Continued)

| | Outcome | | Univariate analysis | Multivariate analysis | | |
|---|---|---|---|---|---|---|
| | Favorable, n = 48 | Unfavorable, n = 45 | p value | odds ratio | 95% CI | p value |
| log E. corrodens, mean±SD | 1.42±0.63 | 1.49±0.58 | 0.59 | | | |

An unfavorable 3-month outcome was defined as a 3 or higher on the modified Rankin Scale.

CI, confidence interval; SD, standard deviation; NIHSS, NIH Stroke Scale; CRP, C-reactive protein.

* means $p < 0.05$.

inflammatory chemokines and cytokines. Together, these reports suggest that *F. nucleatum* may have a systemic adverse impact on stroke patients apart from hemorrhage growth.

There were some limitations to this study. First, there was the issue of sample size and sampling bias. In this study, the sample size was modest. Thus, despite the multicenter study design, sample size calculation, and post hoc analysis, some sampling bias might exist. The average age of patients and the numbers of patients with CAA, previous stroke, and taking antithrombotic drugs were also relatively high, whereas the total frequency of intracerebral hematoma growth was low. Regarding this low intracerebral hematoma growth frequency, the time from onset to admission might have affected the results. In this study, we included patients within 7 days from onset. However, all patients were considered during the acute phase, and the median time from onset to admission was 147 minutes. Time from onset to collection of blood sample also varied because of the study design. However, all patients were tested within a week from onset. Since periodontal disease is a chronic disease, mild variation might not affect the results. Second, this was a cross-sectional observational study, which makes it difficult to adequately assess the biological relationship between periodontal pathogens and cerebral hemorrhage. To provide more conclusive evidence, *in vivo* animal experiments are required. If periodontal pathogens themselves are risk factors for a malignant outcome of cerebral hemorrhage, daily oral care and regular dental examination could practically improve the clinical course. Third, serum IgG titers were used to investigate the association between periodontal disease and the clinical course of a cerebral hemorrhage, but the severity of periodontal disease and the intraoral environment was not directly evaluated. Therefore, future studies could consider these parameters and assess how they affect cerebral hemorrhage pathology and clinical outcomes. While the influence of serum IgG titers of periodontal pathogen on cerebral hemorrhage has not been fully determined, specific periodontal pathogen infection can be a useful biomarker for predicting the clinical course of cerebral hemorrhage.

Stroke mortality rates have been decreasing owing to the development of advanced medical treatments; however, disability rates of stroke survivors have been increasing. Given our findings of an association between periodontal pathogen titers and the clinical course of a cerebral hemorrhage, it is important to seriously consider each of these pathogens and their potential negative impact on affected individuals.

## Conclusions

Our observations reveal the impact of two periodontal pathogens on cerebral hemorrhage, namely that elevated serum IgG titers of *A. actinomycetemcomitans* predicted hemorrhage growth and that those of *F. nucleatum* predicted a poor outcome in patients with this disease. Consequently, periodontal pathogens could play an important role in the management and treatment of stroke patients, for which the assessment of IgG titers has shown to be a valid tool.

## Supporting information

**S1 File. COI.**
(PDF)

**S2 File. All relevant data of the study.**
(XLSX)

## Acknowledgments

We would like to sincerely thank Ms. Akiko Hirata, Ms. Masami Nishino, Ms. Masako Fukuta, and Ms. Kanami Ogawa at the Suiseikai Kajikawa Hospital for their technical assistance.

## Author Contributions

**Conceptualization:** Masahiro Nakamori, Naohisa Hosomi, Hiromi Nishi, Shiro Aoki, Tomohisa Nezu, Yuji Shiga, Naoto Kinoshita, Hiroyuki Kawaguchi, Hidemi Kurihara, Hirofumi Maruyama.

**Data curation:** Masahiro Nakamori, Naohisa Hosomi, Hiromi Nishi, Shiro Aoki, Tomohisa Nezu, Yuji Shiga.

**Formal analysis:** Masahiro Nakamori, Naohisa Hosomi, Hiromi Nishi, Yuji Shiga.

**Funding acquisition:** Naohisa Hosomi.

**Investigation:** Masahiro Nakamori, Naohisa Hosomi, Hiromi Nishi, Yuji Shiga, Naoto Kinoshita, Kenichi Ishikawa, Eiji Imamura, Tomoaki Shintani, Hiroki Ohge, Shinichi Wakabayashi.

**Methodology:** Masahiro Nakamori, Naohisa Hosomi, Hiromi Nishi, Shiro Aoki, Tomohisa Nezu, Yuji Shiga, Kenichi Ishikawa, Eiji Imamura, Tomoaki Shintani, Hiroki Ohge.

**Supervision:** Hiroki Ohge, Hiroyuki Kawaguchi, Hidemi Kurihara, Shinichi Wakabayashi, Hirofumi Maruyama.

**Validation:** Naohisa Hosomi, Hiroki Ohge, Hiroyuki Kawaguchi, Hirofumi Maruyama.

**Writing – original draft:** Masahiro Nakamori, Naohisa Hosomi, Hiromi Nishi, Shiro Aoki, Yuji Shiga.

**Writing – review & editing:** Masahiro Nakamori, Naohisa Hosomi, Shinichi Wakabayashi, Hirofumi Maruyama.

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
