## [Decision Letter · Decision Letter 0]

3 Jun 2020

PONE-D-20-12711

Serum IgG titers of periodontal pathogens predict cerebral hemorrhage growth and 3-month outcome

PLOS ONE

Dear Dr. Hosomi,

Thank you for submitting your manuscript to PLOS ONE. After careful consideration, we feel that it has merit but does not fully meet PLOS ONE’s publication criteria as it currently stands. Therefore, we invite you to submit a revised version of the manuscript that addresses the points raised during the review process.

This is a very interesting manuscript, addressing an uncommon but reasonable association among pathogens and their possible role in hemorrhagic stroke behavior, but many major issues arising from methodology and results, should be analyzed before considering this manuscript as suitable for publication.

We look forward to receiving your revised manuscript.

Kind regards,

Miguel A. Barboza, MD, MSc

Academic Editor

PLOS ONE

Journal Requirements:

"Hirofumi Maruyama reports research support from Eisai, Pfizer, Takeda

Pharmaceutical, Otsuka Pharmaceutical, Nihon Pharmaceutical, Shionogi, Teijin

Pharma, Fuji Film, Boehringer Ingelheim, Sumitomo Dainippon Pharma, Nihon Medi-

Physics, Bayer, MSD, Daiichi Sankyo, Kyowa Hakko Kirin, Sanofi, Novartis, Kowa

Pharmaceutical, Astellas Pharma, Tsumura, Japan Blood Products Organization,

Mitsubishi Tanabe Pharma, Mylan, which are unrelated to the submitted work. All other

authors declare that they have no conflicts of interest."

Reviewers' comments:

Reviewer's Responses to Questions

**Comments to the Author**

1. Is the manuscript technically sound, and do the data support the conclusions?

Reviewer #1: Partly

Reviewer #2: Yes

2. Has the statistical analysis been performed appropriately and rigorously? 

Reviewer #1: Yes

Reviewer #2: I Don't Know

3. Have the authors made all data underlying the findings in their manuscript fully available?

Reviewer #1: No

Reviewer #2: Yes

4. Is the manuscript presented in an intelligible fashion and written in standard English?

Reviewer #1: Yes

Reviewer #2: Yes

5. Review Comments to the Author

Reviewer #1: With glad and interest I have reed the manuscript titled. .Serum IgG titers of periodontal pathogens predict cerebral hemorrhage growth and 3- month outcome

The study is relevant for focusing on five pivotal aspects:

a.- The study on intracerebral hemorrhage (ICH)

b.- The study of periodontal disease

c.- Hematoma growth or hematoma enlargement.

d.- The relationship between them

e.- The short-term outcome

In a prospective (observational cross-sectional) design study from two Japanese hospitals were authors evaluated the growth of hematoma and the short-term outcome in 115 patients with acute ICH and its possible association with the serum IgG titers of 9 of the most frequent periodontal pathogens.

In short, one periodontal pathogen was associated to intracerebraral growth hematoma and anotherone to 3-months poor outcome (A. actinomycetemcomitans and F. nucleatum respectively).

Methods.

Measurement of serum antibody titers of periodontal pathogens serum samples were collected from patients within 3 days after stroke onset.

Q.- Why 3 days after ICH onset?

Results

1.- Low modest sample

2.- Old age average of patients

3.- Low total frequency of intracerebraral growth hematoma (11.3%)

4.- Previos stroke and previos antithrombotic drus in 27.3% and 30.6% respectively

5.- Some patients with 5 days admission after ICH onset symptoms (maximum=7200 minutes) Table 1.

6.- Cerebral amyloid angiopathy 13.9% (definitive, probable or possible?)

Q_ Please, clarify and discuss this findings, especially those that may affect or bias the analysis of the results (hematoma growth and/or outcome)

Also:

1.- If previous stroke (please clarify type: ischemic or hemorrhagic)

2.- Time from previous stroke (please clarify: years or recently in days or months?)

3.- None patients with amyloid angiopathy observed hematoma growth. Please discuss this intrigant finding.

4.- Intraventricular ICH frequency, please clarify

5.- Surgical Management approach, please clarify frequency

6.- Frequency of tube ventilatory use or other orofaringeal devices

7.- In hospital septic complications frequency

Reviewer #2: This article addresses the association between serum IgG titers of periodontal pathogens as predictors for cerebral hemorrhage growth and poor functional outcome at 3 months (i.e. modified Ranking Scale �3). The topic is interesting and it has not been explored before on a prospective cross-sectional study for cerebral hemorrhage. The abstract is appropriate for the content of the text. The authors mentioned relevant information regarding the ‘non-availability’ for reversal agents for oral anticoagulants by the time the study underwent, and their awareness for lack of periodontal disease clinical status from participants. Although the analysis included the most common vascular risk factors, there are some clinical characteristics that should have been considered to adjust during the statistical analysis, such as: Glasgow Coma Scale score, intraventricular hemorrhage extension, and early hematoma growth neuroimaging predictors (e.g. blend sign). Furthermore, since early blood pressure control in acute intracerebral hemorrhage is a key prognostic factor, it would have been useful to be mentioned if any of the participants required pharmacological blood pressure management during the first 24 hours, and consider to adjust for this on the statistical analysis. The conclusions that “elevated serum IgG titers of Aggregatibacter actinomycetemcomitans is associated to cerebral hemorrhage growth at 24 hours follow-up from admission and yet no effect on functional outcome at 3-months follow-up, and that elevated IgG titers of Fusobacterium nucleatum predicted a poor outcome”, are reasonable given the data reported in the article, but overreached the collected data.

Please find attached my additional comments for the authors.

Best wishes.

6. PLOS authors have the option to publish the peer review history of their article (what does this mean?). If published, this will include your full peer review and any attached files.

Reviewer #1: No

Reviewer #2: No

---

## [Author Response · Author response to Decision Letter 0]

26 Aug 2020

Thank you again for reviewing our manuscript. We appreciate the insightful comments and advice of the reviewers. We have provided point-by-point responses to each of the comments below and highlighted the corresponding revisions in the manuscript document in red.

Response to the editor

Comment 1: Please ensure that your manuscript meets PLOS ONE's style requirements, including those for file naming. The PLOS ONE style templates can be found at

Response 1: We will ensure and meet the PLOS ONE's style requirements.

Comment 2: Thank you for stating the following in the Competing Interests section:

"Hirofumi Maruyama reports research support from Eisai, Pfizer, Takeda

Pharmaceutical, Otsuka Pharmaceutical, Nihon Pharmaceutical, Shionogi, Teijin

Pharma, Fuji Film, Boehringer Ingelheim, Sumitomo Dainippon Pharma, Nihon Medi-

Physics, Bayer, MSD, Daiichi Sankyo, Kyowa Hakko Kirin, Sanofi, Novartis, Kowa

Pharmaceutical, Astellas Pharma, Tsumura, Japan Blood Products Organization,

Mitsubishi Tanabe Pharma, Mylan, which are unrelated to the submitted work. All other

authors declare that they have no conflicts of interest."

Please confirm that this does not alter your adherence to all PLOS ONE policies on sharing data and materials, by including the following statement: "This does not alter our adherence to PLOS ONE policies on sharing data and materials.” (as detailed online in our guide for authors http://journals.plos.org/plosone/s/competing-interests). If there are restrictions on sharing of data and/or materials, please state these. Please note that we cannot proceed with consideration of your article until this information has been declared. Please include your updated Competing Interests statement in your cover letter; we will change the online submission form on your behalf. Please know it is PLOS ONE policy for corresponding authors to declare, on behalf of all authors, all potential competing interests for the purposes of transparency. PLOS defines a competing interest as anything that interferes with, or could reasonably be perceived as interfering with, the full and objective presentation, peer review, editorial decision-making, or publication of research or non-research articles submitted to one of the journals. Competing interests can be financial or non-financial, professional, or personal. Competing interests can arise in relationship to an organization or another person. Please follow this link to our website for more details on competing interests: http://journals.plos.org/plosone/s/competing-interests

Response 2: We added the following sentence in the Competing Interests section and included the Competing Interests statement in the cover letter:

This does not alter our adherence to PLOS ONE policies on sharing data and materials.

 

Response to the reviewers

Reviewer #1: With glad and interest I have read the manuscript titled. Serum IgG titers of periodontal pathogens predict cerebral hemorrhage growth and 3- month outcome. The study is relevant for focusing on five pivotal aspects:

a.- The study on intracerebral hemorrhage (ICH)

b.- The study of periodontal disease

c.- Hematoma growth or hematoma enlargement.

d.- The relationship between them

e.- The short-term outcome

In a prospective (observational cross-sectional) design study from two Japanese hospitals were authors evaluated the growth of hematoma and the short-term outcome in 115 patients with acute ICH and its possible association with the serum IgG titers of 9 of the most frequent periodontal pathogens.

In short, one periodontal pathogen was associated to intracerebraral growth hematoma and anotherone to 3-months poor outcome (A. actinomycetemcomitans and F. nucleatum respectively).

Response: We appreciate your review and suggestions. To prevent as much bias as possible, we have added the clinical information which the reviewers suggested and analyzed it. In addition, we added the sample size calculation, statistical power, significance level, and effect size for statistical confirmation. However, there were some sampling biases, which we have mentioned as one of the limitations of the study.

Comment 1:

Methods.

Measurement of serum antibody titers of periodontal pathogens serum samples were collected from patients within 3 days after stroke onset.

Q.- Why 3 days after ICH onset?

Response 1: We tried to collect the serum samples as soon as possible. However, it took time to obtain the consent from the patients or their relatives. In this way, we decided on a time within 3 days due to the procedural issue. All patients were tested within a week from onset. Since periodontal disease is a chronic disease, mild variations in the sampling time might not affect the results. We added the following text to the ‘Discussion’ section:

Page 26, Lines 288-291

Time from onset to collection of blood sample also varied because of the study design. However, all patients were tested within a week from onset. Since periodontal disease is a chronic disease, mild variation might not affect the results.

Comment 2:

Results

1.- Low modest sample

2.- Old age average of patients

3.- Low total frequency of intracerebral growth hematoma (11.3%)

4.- Previous stroke and previous antithrombotic drugs in 27.3% and 30.6% respectively

5.- Some patients with 5 days admission after ICH onset symptoms (maximum=7200 minutes) Table 1.

6.- Cerebral amyloid angiopathy 13.9% (definitive, probable or possible?)

Q_ Please, clarify and discuss this findings, especially those that may affect or bias the analysis of the results (hematoma growth and/or outcome)

Response 2: We appreciate your suggestions. In this study, the sample size was modest. Thus, despite the multicenter study design, sample size calculation, and post hoc analysis, some sampling bias might exist. As the reviewer mentioned, the average age of patients was high. The numbers of patients with CAA (7 patients were probable CAA and 6 patients were possible CAA), previous stroke, and taking antithrombotic drugs were also relatively high, whereas the total frequency of intracerebral hematoma growth was low. Regarding this low intracerebral hematoma growth frequency, the time from onset to admission might have affected the results. In this study, we included patients within 7 days from onset. As the reviewer pointed, one patient was admitted 5 days after intracerebral hemorrhage onset. Especially for patients with CAA, admission after onset tended to be delayed, because the symptoms were mild regardless of the bleeding. We understand that time from onset to admission affects hematoma growth. However, all patients were considered during the acute phase, and the median time from onset to admission was 137 minutes. To address these issues, we have modified the Materials and Methods and Results sections and discussed the possible sampling biases as limitations in the Discussion. We also added the relevant clinical information in Tables 1, 3, and 4.

Page 8, Lines 107-110

Conscious level was evaluated with Glasgow coma scale. Comorbidities were defined according to a previous report [13] based on the Japanese hypertension, diabetes mellitus, dyslipidemia, atrial fibrillation, and chronic kidney disease guidelines.

Page 9, Lines 130-137

The other two experienced neurologists (NK and KI) also performed evaluations of MRI findings and hematoma growth neuroimaging predictors, which were detected as the blend sign and black hole sign using plain head CT [15]. Cerebral amyloid angiopathy (CAA) was diagnosed using modified Boston criteria [16]. 

We collected clinical data regarding the acute phase, including intraventricular hemorrhage extension, pharmacological blood pressure management during the first 24 hours, surgical management approach, tube ventilatory use, and septic complications. 

Page 12, Lines 173-176

Because there were no reports of hemorrhagic stroke, we calculated the sample size according to the past investigations for the IgG titers of periodontal pathogens in atherothrombotic stroke [9]. Based on an alpha level = 0.05 and power = 0.80, we estimated that we would require a total of n = 99 participants.

Page 12, Lines 181-182

We considered p < 0.05 as statistically significant. We also calculated the statistical power and effect size as post hoc analysis.

Pages 12-13, Lines 187-191

Among them, 32 (27.8%) patients had histories of stroke, 16 of whom were cerebral hemorrhage. The number of patients with anti-platelet and anticoagulant drug use was 20 (17.4%) and 15 (13.2%), respectively. The median time from onset to admission was 147 minutes (min–max: 26–7200). CAA was diagnosed 16 (13.9%) patients, of whom 7 patients were probable CAA and 6 patients were possible CAA.

Pages 15-16, Lines 210-216

Multivariate logistic regression analysis revealed that usage of anticoagulant (odds ratio 8.36, 95% CI 1.35–51.70, p = 0.02), septic complications (odds ratio 10.20, 95% CI 1.94–53.72, p = 0.01), and the IgG titer of A. actinomycetemcomitans (odds ratio 5.26, 95% CI 1.52–18.25, p = 0.01) were independently associated with hemorrhage growth (Table 3). Regarding the IgG titer of A. actinomycetemcomitans, the statistical power and effect size of Cohen’s d were 0.80 and 0.67, respectively.

Page 20, Lines 227-233

Multivariate logistic regression analysis revealed that age (odds ratio 1.09, 95% CI 1.01–1.17, p = 0.02), NIHSS score (odds ratio 1.29, 95% CI 1.09–1.52, p = 0.002), and the IgG titer of F. nucleatum (odds ratio 7.86, 95% CI 1.08–57.08, p = 0.04) were independently associated with poor outcome, but not cerebral hematoma volume or growth (Table 4). Regarding the IgG titer of F. nucleatum, the statistical power and effect size of Cohen’s d were 0.91 and 0.69, respectively.

Page 26, Lines 279-291

There were some limitations to this study. First, there was the issue of sample size and sampling bias. In this study, the sample size was modest. Thus, despite the multicenter study design, sample size calculation, and post hoc analysis, some sampling bias might exist. The average age of patients and the numbers of patients with CAA, previous stroke, and taking antithrombotic drugs were also relatively high, whereas the total frequency of intracerebral hematoma growth was low. Regarding this low intracerebral hematoma growth frequency, the time from onset to admission might have affected the results. In this study, we included patients within 7 days from onset. However, all patients were considered during the acute phase, and the median time from onset to admission was 147 minutes. Time from onset to collection of blood sample also varied because of the study design. However, all patients were tested within a week from onset. Since periodontal disease is a chronic disease, mild variation might not affect the results.

Comment 3:

Also:

1.- If previous stroke (please clarify type: ischemic or hemorrhagic)

2.- Time from previous stroke (please clarify: years or recently in days or months?)

3.- None patients with amyloid angiopathy observed hematoma growth. Please discuss this intrigant finding.

4.- Intraventricular ICH frequency, please clarify

5.- Surgical Management approach, please clarify frequency

6.- Frequency of tube ventilatory use or other orofaringeal devices

7.- In hospital septic complications frequency

Response 3: We appreciate your suggestions. To prevent as much bias as possible, we have added the clinical information which the reviewers suggested and analyzed it. We added the information of previous stroke (ischemic or hemorrhagic), time from previous stroke, intraventricular ICH frequency, surgical management approach, frequency of tube ventilatory use, and septic complications frequency. Indeed, it was intriguing that none of the patients with amyloid angiopathy had any hematoma growth. One possible reason for this is that in the patients with CAA, admission after onset tended to be delayed, because the symptoms were mild regardless of the bleeding. Sampling bias might also affect the results. Thus, we described the sampling biases as limitations in the ‘Discussion’ section (the changes were described as the previous response). We have also revised Tables 1, 3, and 4 accordingly.

 

Reviewer #2:

This article addresses the association between serum IgG titers of periodontal pathogens as predictors for cerebral hemorrhage growth and poor functional outcome at 3 months (i.e. modified Ranking Scale �3). The topic is interesting and it has not been explored before on a prospective cross-sectional study for cerebral hemorrhage. The abstract is appropriate for the content of the text. The authors mentioned relevant information regarding the ‘non-availability’ for reversal agents for oral anticoagulants by the time the study underwent, and their awareness for lack of periodontal disease clinical status from participants. Although the analysis included the most common vascular risk factors, there are some clinical characteristics that should have been considered to adjust during the statistical analysis, such as: Glasgow Coma Scale score, intraventricular hemorrhage extension, and early hematoma growth neuroimaging predictors (e.g. blend sign). Furthermore, since early blood pressure control in acute intracerebral hemorrhage is a key prognostic factor, it would have been useful to be mentioned if any of the participants required pharmacological blood pressure management during the first 24 hours, and consider to adjust for this on the statistical analysis. The conclusions that “elevated serum IgG titers of Aggregatibacter actinomycetemcomitans is associated to cerebral hemorrhage growth at 24 hours follow-up from admission and yet no effect on functional outcome at 3-months follow-up, and that elevated IgG titers of Fusobacterium nucleatum predicted a poor outcome”, are reasonable given the data reported in the article, but overreached the collected data. 

Below there are more specific comments that should be clarified/addressed: 

Response: We appreciate for your review and suggestions. To prevent the bias as possible, we included the clinical information such as Glasgow Coma Scale score, intraventricular hemorrhage extension, early hematoma growth neuroimaging predictors, if any of the pharmacological blood pressure management during the first 24 hours, time from previous stroke, surgical management approach, tube ventilatory use, and septic complication. In addition, we added the sample size calculation, statistical power, significance level and effect size for statistical confirmation.

Major issues: 

Comment 1: P. 10: Please provide prevalence estimates regarding the IgG titers of periodontal pathogens in hemorrhagic stroke taken from the literature. More information needs to be added about the sample size calculation, statistical power, significance level and effect size. The sample size needs to be more extensively explained by the authors; this will allow to be precise about sizes effect (if any) in order to make solid conclusions and give opportunity to generate new hypothesis for future research.

Response 1: Because there were no reports regarding to the hemorrhagic stroke, we calculated the sample size according to the past investigations for IgG titers of periodontal pathogens in atherothrombotic stroke. We have added the statistical power, significance level, and effect size to the revised the manuscript and added statements regarding the IgG titers as follows:

Page 12, Lines 173-182

Because there were no reports of hemorrhagic stroke, we calculated the sample size according to the past investigations for the IgG titers of periodontal pathogens in atherothrombotic stroke [9]. Based on an alpha level = 0.05 and power = 0.80, we estimated that we would require a total of n = 99 participants. Baseline data of cerebral hemorrhage patients were analyzed, and two-step strategies were employed to assess the relative importance of variables in their association with hemorrhage growth and poor outcome using least square linear regression analysis. We first performed a univariate analysis, followed by a multi-factorial analysis with selected factors with p < 0.05 in the former analysis. We considered p < 0.05 as statistically significant. We also calculated the statistical power and effect size as post hoc analysis.

Pages 15-16, Lines 210-216

Multivariate logistic regression analysis revealed that usage of anticoagulant (odds ratio 8.36, 95% CI 1.35–51.70, p = 0.02), septic complications (odds ratio 10.20, 95% CI 1.94–53.72, p = 0.01), and the IgG titer of A. actinomycetemcomitans (odds ratio 5.26, 95% CI 1.52–18.25, p = 0.01) were independently associated with hemorrhage growth (Table 3). Regarding the IgG titer of A. actinomycetemcomitans, the statistical power and effect size of Cohen’s d were 0.80 and 0.67, respectively.

Page 20, Lines 227-233

Multivariate logistic regression analysis revealed that age (odds ratio 1.09, 95% CI 1.01–1.17, p = 0.02), NIHSS score (odds ratio 1.29, 95% CI 1.09–1.52, p = 0.002), and the IgG titer of F. nucleatum (odds ratio 7.86, 95% CI 1.08–57.08, p = 0.04) were independently associated with poor outcome, but not cerebral hematoma volume or growth (Table 4). Regarding the IgG titer of F. nucleatum, the statistical power and effect size of Cohen’s d were 0.91 and 0.69, respectively.

Comment 2: P. 12: Table 2. Select a more concise heading. It is not clear if the serum IgG titers makes reference to the ‘reference range’ from the five healthy volunteers or if these measurements are considered significant positive titers. It could be helpful to add a brief explanation on the ‘Results’ section. 

Response 2: In Table 2, we show the mean ± SD of the serum IgG titers from all subjects in the study. As mentioned in the ‘Materials and Methods’ section, the standard reaction was defined based on the ELISA unit (EU) such that 100 EU corresponded to a 1:3200 dilution of the calibrator sample (5 healthy controls). Because there are no defined positive cutoff values, we reported the mean ± SD of the serum IgG titers from all subjects in the study. We have rewritten the heading and part of the ‘Results’ section.

Page 13, Lines 192-193.

The mean serum IgG titers of periodontal disease pathogens from all patients are summarized in Table 2.

Comment 3: P. 14: Table 3. It would be more informative if you add to the ‘P values’ the odds ratio and confidence intervals for the univariate and multivariable analysis. Also, notice that data from the table considered four patients with history of atrial fibrillation and five patients taking anticoagulants on the positive hematoma growth group. On this same group, from 13 patients with positive hematoma growth, twelve had a hypertensive etiology, zero amyloid angiopathy and the remaining one participant is not mentioned on the table data. It is relevant that authors address this on the results section and verified data in this table. 

Response 3: We have added the odds ratio and confidence intervals into Table 3. One patient without atrial fibrillation took anticoagulants because of risk of cardioembolism due to left ventricular dysfunction. There were some cerebral hemorrhage patients with other etiology such as arteriovenous malformation and moyamoya disease. We also added a row for other etiologies in Table 3. 

Comment 4: P. 17: Table 4. It would be more informative if you add to the ‘P values’ the odds ratio and confidence intervals for the univariate and multivariable analysis.

Response 4: In accordance with your suggestion, we have incorporated the odds ratio and confidence intervals into Table 4.

Minor issues: 

1. Introduction: 

Comment 5: P. 5: Please add more background information about the potential mechanism proposed for the influence of periodontal disease in stroke since this will give more context to the reader. It is required more detail about the large cohort study that demonstrated the association between periodontal disease and the incidence of specific etiologies for stroke (i.e. include hazard ratio and confidence interval for these statements). 

Response 5: We appreciate your suggestion. We have added more background information about the influence of periodontal disease in stroke as follows:

Pages 5-6, Lines 58-77

Meta-analysis from previous studies showed that the risk of stroke was significantly increased in individuals with periodontitis in which the relative risk was 1.63 (95% confidence interval [CI] 1.25–2.00) [5]. Furthermore, a large cohort study has demonstrated that periodontal disease is associated with the incidence of cardioembolic and atherothrombotic stroke, and that regular dental care might decrease stroke risk [6]. Periodontitis is related to an increase in systemic inflammation markers through exposure to Gram-negative bacteria, which are implicated in the etiology of stroke [7]. While the treatment of stroke has improved remarkably, the management of periodontal disease may not only improve the clinical course in patients but could help prevent stroke altogether. However, in these studies periodontal diseases were diagnosed by only oral examination.

 It was reported that serum IgG titers of a certain periodontal pathogen are considered to reflect its periodontal status [8]. Recently, by analyzing serum antibody titers, certain periodontal pathogens have been identified as risk factors for systemic diseases, such as ischemic stroke, coronary heart disease, non-alcoholic fatty liver disease, and Alzheimer's disease [9-12]. Especially regarding ischemic stroke, the serum antibody level of Prevotella (P.) intermedia was significantly higher in atherothrombotic stroke patients than in patients with no previous stroke [9]. However, there is no reported association between the serum IgG titers of periodontal pathogens and cerebral hemorrhage.

Comment 6: P. 6: Regarding the aim of the study at the end of the introduction – ‘Therefore, to understand how’ – could be rephrase for – ‘to determine if’ – since the outcome is about prognosis and not pathophysiology. 

Response 6: We have revised the manuscript as per your suggestion.

Page 6, Lines 80-82

To determine if periodontal disease affects cerebral hemorrhage, we examined the relationship among serum IgG titers of periodontal pathogens, cerebral hemorrhage growth, and a 3-month outcome.

2. Materials and methods: 

Comment 7: P. 6: It is not clear all the inclusion and exclusion criteria for participants (e.g. >18 years-old, etiologies stated as ‘others’ cerebral hemorrhage causes). 

Response 7: We have revised the section on the enrollment criteria as follows:

Page 7, Lines 94-99

We included patients who were admitted within 7 days from onset, were aged ≥ 20 years, and for whom consent to participate in this study was obtained from the patient or their relatives. We excluded patients who could not undergo head computed tomography (CT) and magnetic resonance imaging (MRI). We also excluded the patients who were diagnosed with hemorrhagic infarction or trauma-induced hemorrhages.

Comment 8: P. 7: Please confirm if participants underwent head computed tomography and/or magnetic resonance at admission, and please clarify who performed the NIHSS assessment. 

Response 8: Imaging analysis with head computed tomography (CT) and magnetic resonance imaging was performed in all patients for acute cerebral hemorrhage diagnosis. Accordingly, we added the following text:

Page 8, Lines 105-106

Two stroke specialists (SA and EI) evaluated the stroke severity and conscious level.

Comment 9: P. 7-8: It is worth to include the references for previously established definitions such as: ‘hypertension’ (e.g. American or European Hypertension Guidelines), ‘diabetes mellitus’, ‘dyslipidemia’, and ‘atrial fibrillation’; as there could be small differences on diagnostic criteria. 

Response 9: We appreciate your wise advice. We have added citations and references to the appropriate reports and guidelines.

Page 8, Lines 107-110

Conscious level was evaluated with Glasgow coma scale. Comorbidities were defined according to a previous report [13] based on the Japanese hypertension, diabetes mellitus, dyslipidemia, atrial fibrillation, and chronic kidney disease guidelines.

Comment 10: P. 8: It is worth to explicitly state if the neuroimaging evaluation (i.e. hematoma volume measurements) remained blinded from the clinical assessment. 

Response 10: We have added the following text to the revised manuscript:

Page 9, Line 128

The neuroimaging evaluation remained blinded from the clinical assessment.

Comment 11: P. 9: Regarding the determination of serum IgG titers, more detail about sampling and processing methods would be welcomed or they could be revisited as supplementary material. 

Response 11: We have added a detailed method to the revised manuscript as follows:

Page 10, Lines 144-153

Bacterial antigen-coated wells were washed with phosphate-buffered saline with Tween (PBST); serum samples in PBST were then added to the wells. After incubation at 4°C overnight, the wells were washed with PBST and filled with alkaline phosphatase-conjugated goat anti-human IgG (gamma-chain specific, Abcam, Cambridge, MO) in PBST. After another incubation at 37°C for 2 hours, the wells were again washed with PBST, an aliquot of p-nitrophenylphosphate at 1 mg/mL (WAKO Pure Chemical Industries Ltd., Osaka, Japan) in 10% diethanolamine buffer was added to each well as a substrate, and incubation was performed at 37°C for 30 minutes. Optical density at 405 nm was measured using a microplate reader (iMark, Bio-Rad Laboratories Inc., Hercules, CA).

3. Discussion

Comment 12: P. 16, line 3: It is recommended to unified the format ‘(listed in Tables 1 & 2, and cerebral hematoma growth)’ instead it could be referred as ‘(listed in Tables. 1, 2 and 3)’.

Response 12: We have revised the format as you have suggested.

Comment 13: P. 16: Limitations should be updated, including comments regarding sample size and others addressed at the beginning of this review. 

Response 13: We appreciate your suggestions. We had calculated the sample size and performed post hoc analysis. In addition, this study was performed by multicenter. However, there might exist some sampling bias as the reviewers mentioned. We described and added these sampling bias as limitations in the ‘Discussion’ section. 

Page 26, Lines 279-291

There were some limitations to this study. First, there was the issue of sample size and sampling bias. In this study, the sample size was modest. Thus, despite the multicenter study design, sample size calculation, and post hoc analysis, some sampling bias might exist. The average age of patients and the numbers of patients with CAA, previous stroke, and taking antithrombotic drugs were also relatively high, whereas the total frequency of intracerebral hematoma growth was low. Regarding this low intracerebral hematoma growth frequency, the time from onset to admission might have affected the results. In this study, we included patients within 7 days from onset. However, all patients were considered during the acute phase, and the median time from onset to admission was 147 minutes. Time from onset to collection of blood sample also varied because of the study design. However, all patients were tested within a week from onset. Since periodontal disease is a chronic disease, mild variation might not affect the results.

4. Title

Comment 14: P. 1: Consider using a more accurate term such as ‘association’ or ‘relationship’ instead of ‘predict’. This would be more appropriate for the reader expectations. 

Response 14: We have changed the title as per your suggestion.

---

## [Decision Letter · Decision Letter 1]

12 Oct 2020

Serum IgG titers against periodontal pathogens are associated with cerebral hemorrhage growth and 3-month outcome

PONE-D-20-12711R1

Dear Dr. Hosomi,

We’re pleased to inform you that your manuscript has been judged scientifically suitable for publication and will be formally accepted for publication once it meets all outstanding technical requirements.

Kind regards,

Miguel A. Barboza, MD, MSc

Academic Editor

PLOS ONE

Additional Editor Comments (optional):

Reviewers' comments:

Reviewer's Responses to Questions

**Comments to the Author**

1. If the authors have adequately addressed your comments raised in a previous round of review and you feel that this manuscript is now acceptable for publication, you may indicate that here to bypass the “Comments to the Author” section, enter your conflict of interest statement in the “Confidential to Editor” section, and submit your "Accept" recommendation.

Reviewer #1: All comments have been addressed

Reviewer #2: All comments have been addressed

2. Is the manuscript technically sound, and do the data support the conclusions?

Reviewer #1: Yes

Reviewer #2: Yes

3. Has the statistical analysis been performed appropriately and rigorously? 

Reviewer #1: Yes

Reviewer #2: Yes

4. Have the authors made all data underlying the findings in their manuscript fully available?

Reviewer #1: No

Reviewer #2: Yes

5. Is the manuscript presented in an intelligible fashion and written in standard English?

Reviewer #1: Yes

Reviewer #2: Yes

6. Review Comments to the Author

Reviewer #1: Authors have clarified the doubts in a timely manner. Comments have been answered as far as possible.

Reviewer #2: I was delighted to review the manuscript. It addresses an interesting major topic (i.e. intracerebral haemorrhage) from a non-conventional and deeply explored perspective (i.e. periodontal pathogens) and it gives place to potential future. The authors reasonably addressed prior comments to their paper. The manuscript is now easy to follow and the results and conclusions are coherent with the methodology. Finally, I would strongly suggest to the authors to continue exploring in further research this interesting association in a higher selective population with more rigorous inclusion and exclusion criteria.

7. PLOS authors have the option to publish the peer review history of their article (what does this mean?). If published, this will include your full peer review and any attached files.

Reviewer #1: No

Reviewer #2: No

---

## [Editor Report · Acceptance letter]

16 Oct 2020

PONE-D-20-12711R1 

Serum IgG titers against periodontal pathogens are associated with cerebral hemorrhage growth and 3-month outcome 

Dear Dr. Hosomi:

I'm pleased to inform you that your manuscript has been deemed suitable for publication in PLOS ONE. Congratulations! Your manuscript is now with our production department. 

Kind regards, 

on behalf of

Dr. Miguel A. Barboza 

Academic Editor

PLOS ONE